# Stiffness Analysis of Parallel Cable-Driven Upper Limb Rehabilitation Robot

**DOI:** 10.3390/mi13020253

**Published:** 2022-02-02

**Authors:** Yupeng Zou, Xiangshu Wu, Baolong Zhang, Qiang Zhang, Andong Zhang, Tao Qin

**Affiliations:** 1College of Mechanical and Electronic Engineering, China University of Petroleum (East China), Qingdao 266580, China; zouyupeng@upc.edu.cn (Y.Z.); s20040023@s.upc.edu.cn (X.W.); s19040027@upc.edu.cn (B.Z.); s19040017@s.upc.edu.cn (Q.Z.); s20040049@s.upc.edu.cn (A.Z.); 2Xiangyang Key Laboratory of Rehabilitation Medicine and Rehabilitation Engineering Technology, Hubei University of Arts and Science, Xiangyang 441053, China; 3School of Mechanical Engineering, Hubei University of Arts and Science, Xiangyang 441053, China

**Keywords:** upper limb rehabilitation robot, cable-driven robot, static stiffness derivation, multi-body simulation analysis

## Abstract

This paper studies the stiffness of the parallel cable-driven upper limb rehabilitation robot (PCUR). Firstly, it was derived that the static stiffness expression of the PCUR was composed of platform pose stiffness ***K***_T_ and cable pose stiffness ***K***_S_. It indicated that the static stiffness of the PCUR was related to the cable tension, the arrangement of the cable, and the cable stiffness. Secondly, a simulation model in MATLAB/Simscape Multibody was built. Cable tension was applied to make the moving platform in a static equilibrium state. The stiffness of the PCUR and the external force on the moving platform were changed, and the motion characteristics of the moving platform were obtained. Finally, the position changes of the moving platform under different external forces were analyzed, and the motion laws of the moving platform under different stiffnesses were summarized.

## 1. Introduction

Stroke has become an important issue of global concern due to its high incidence and high disability rate. Various complications such as hemiplegia caused by it have attracted attention from various fields [1,2,3]. In medical rehabilitation, rehabilitation robots can better perform rehabilitation tasks in place of doctors, which is conducive to patients’ early recovery. The cable used in rehabilitation robots has lower quality and rigidity and is superior to traditional rigid rehabilitation robots in terms of safety and flexibility control [4,5,6]. Therefore, it has become a research focus in the field of medical rehabilitation in recent years. It is important to research the kinematics analysis, dynamics analysis, and overall control strategy of the cable-driven rehabilitation robot.

Cable-driven robots can be divided into two categories—tandem and parallel robots, while parallel robots can be further divided into suspension and constrained robots. Mao and Agrawal proposed a cable-driven arm exoskeleton robot (CAREX), which is part of a tandem mechanism. The CAREX, driven by multiple cables together, solves the problem of a single cable which does not provide thrust [7]. Jin et al. designed a three-degree-of-freedom parallel cable-driven robot for upper limb rehabilitation on a flat surface [8]. Gallina presented the NeRebot robot, with three degrees of freedom of movement, which is a parallel suspension robot [9]. The rehabilitation robot STRING-MAN, designed by Surdilovic, balances the weight of the human body and is a constrained parallel robot [10]. Compared to the mechanisms described above, the mechanism designed in this paper has six degrees of freedom in space, allowing for a more comprehensive rehabilitation movement, and is a constrained parallel robot. The overall size of the robot allows the patient not to be confined to a fixed position, and the increased workspace facilitates the planning of various rehabilitation trajectories.

Research on the stiffness control and force control of the parallel cable-driven upper limb rehabilitation robot (PCUR) can make an overall stiffness adjustment and realize an overall flexibility and a better man–machine interaction of the robot. The cable tension and the combined force of the cable acting on the moving platform are controlled. It can prevent secondary injuries to affected limbs caused by excessive force during rehabilitation training, effectively improve the overall flexibility of the rehabilitation robot and achieve safer and more comfortable rehabilitation training [11,12]. Bruckmann studied the dynamics and stiffness space properties of the cable robot [13,14]. Verhoeven gave the stiffness of the Stewart platform manipulator driven by a cable, indicating that the manipulator’s stiffness is related to the cable stiffness and the pose of the moving platform [15,16]. Albus and Dagalakis analyzed the stiffness characteristics of the crane robot and derived linear and non-linear mathematical models of the stiffness matrix of the end effector, showing that the stiffness is not only related to the stiffness of the cable and the pose of the end effector but also related to the weight of the suspension [17]. Kawamura conducted a stiffness analysis study on the parallel cable high-speed mobile robot FALCON-7 and showed that increasing the cable tension can improve the structural stiffness [18]. Further research on the theoretical analysis of the stiffness of the parallel cable robot has been performed by Cui et al. and Wang et al. [19,20]. Zeng et al. proposed a dynamical movement primitive (DMP) based motion trajectory model and biomimetic adaptive control strategy, which enables robots to learn compliant motor skills and realize variable stiffness control [21].

Patients in different rehabilitation stages need different rehabilitation training modes, mainly divided into the following three modes, i.e., the passive training mode, the active training mode, and the impedance training mode [22,23]. The three rehabilitation training modes have different requirements for the stiffness of the rehabilitation robot [24]. It is of great significance to study the stiffness of the PCUR.

Most previous papers have investigated the stiffness properties of cable robots at a theoretical level, and there is less literature on the stiffness properties associated with actual models. This paper analyzes the stiffness theory of the PCUR and uses it for different stiffness performance analyses, and it was derived that static stiffness ***K*** is composed of two parts: ***K***_T_ is the static stiffness related to the tension of the cable, which is the leading cause of the change in the pose of the moving platform and is called the platform pose stiffness; ***K***_S_ is the static stiffness related to the structural parameters of the robot, which is produced by the change of the cable configuration and is called the cable pose stiffness. By controlling ***K***_T_, rehabilitation can be performed at different stiffnesses. The novelty of this paper is that by building a multi-body simulation model, cable tension required to maintain a certain stiffness of the moving platform was analyzed, and different external forces were applied on the moving platform. Different stiffness conditions were combined to study the movement of the moving platform.

The rest of this paper is organized as follows: The second part gives a brief overview of the PCUR structure. The third part carries on the mathematical modeling and the stiffness theory analysis of the PCUR. The fourth part builds the simulation model, and the range of the stiffness is solved according to the cable tension condition. The fifth part analyzes the simulation results and studies the position changes of the moving platform under different external forces and different stiffnesses. The sixth part is the summary and the prospect of the PCUR static stiffness research.

## 2. Cable-Driven Upper Limb Rehabilitation Robot

The schematic of the PCUR is shown in Figure 1. The PCUR consists of seven cable-driven units to move a retractable cable and traction a moving platform to drive the affected limb to perform rehabilitation actions. The patient sits on a rotatable seat, the seat height is adjustable, and the patient’s posture can be adjusted according to different rehabilitation modes. The PCUR’s moving platform can perform six degrees of freedom (DOF) movements in space. It is a redundant drive mechanism. The PCUR has a simple configuration, an ample working space, and adjustable stiffness, which can meet different upper limb rehabilitation training modes [25,26]. Combined with the control systems design, the stiffness of planning can achieve a more secure and comfortable rehabilitation training, adapt to rehabilitation patients in different rehabilitation training periods and formulate different rehabilitation training plans.

The upper limb rehabilitation training in the form of the robot is shown in Figure 2. The upper limb rehabilitation training includes the upper limb–shoulder joint extension, the lateral extension, the shoulder–elbow joint flexion, and the internal/external rotation training. The upper limb–shoulder joint extension training moves in extension in the XOZ plane and rotates around the Y-axis. The upper limb–shoulder joint lateral extension training is within the YOZ plane and with rotation around the X-axis. The upper limb–shoulder–elbow joint extension training is performed in the XOY plane, rotating around the Z-axis. The upper limb–shoulder–elbow joint internal/external rotation training is the rotation about the X-axis. Different training modes need to achieve the best training effect under proper cable tension and stiffness conditions. It also ensures the safety and comfort of upper limb rehabilitation training.

Based on the PCUR, the static analysis and stiffness analysis of the mechanism was carried out. The simulation model in MATLAB/Simscape Multibody was built, and the motions of the moving platform under external forces were studied. The performances of PCUR under different stiffnesses and the appropriate stiffness of rehabilitation training were analyzed, laying the foundation for the follow-up control of the PCUR’s stiffness.

## 3. Modeling and Static Stiffness Analysis

### 3.1. Static Analysis

As shown in Figure 3, the mathematical modeling and analysis of PCUR were carried out. B1–B7 represents the cable driving points. P1–P7 represents the traction points of the moving platform. The center point of the moving platform in the static state coincides with the origin of the global coordinate system *O-xyz*. The local coordinate system *P-x′y′z′* represents the center point position of the moving platform.

The pose of the moving platform in the global coordinate system is expressed as:(1)$$S={\left[\begin{array}{cccccc} x & y & z & \alpha & \beta & \gamma \end{array}\right]}^{T}.$$

The unit vector $u_{i}$ in the cable direction is expressed as:(2)$$u_{i}=\frac{L_{i}}{l_{i}},$$
where ***L****_i_* represents the vector of cable length and *l_i_* represents the cable length.

***r****_i_* represents the vector from the center point of the moving platform to the cable traction point and is described as:(3)ri=R⋅PPi,
where PPi represents the coordinates of the *i*-th cable connection point $P_{i}$ in the local coordinate system *P-x′y′z′*; ***R*** is the rotation matrix of the local coordinate system *P-x′y′z′* relative to the global coordinate system *O-xyz*. Abbreviating *cos* to *c* and *sin* to *s* for the simplicity of writing, the expression of ***R*** is written as:(4)$$R=\left[\begin{array}{ccc} c\gamma c\beta & c\gamma s\beta s\alpha -s\gamma c\alpha & c\gamma s\beta c\alpha +s\gamma s\alpha \\ s\gamma c\beta & s\gamma s\beta s\alpha +c\gamma c\alpha & s\gamma s\beta c\alpha -c\gamma s\alpha \\ -s\beta & c\beta s\alpha & c\beta c\alpha \end{array}\right].$$

The PCUR is a redundant mechanism (*m* represents the number of cables, and *n* represents the degree of freedom of the mechanism; *m* > *n*), which can realize the motion in the direction of six degrees of freedom in space. The moving platform maintains equilibrium under the joint action of seven cables. The static equilibrium expression is shown as following:(5)$$J^{T}T+F=0,$$
where ***J***^T^ is the system structure matrix, ***T*** represents the tension vector of the cable, and ***F*** represents the generalized external force of the moving platform, including its gravity and the inertial force and the centrifugal force in the process of moving. The specific expression of the structure matrix ***J***^T^ is written as:(6)$$J^{T}=\left[\begin{array}{cccc} u_{1} & u_{2} & \cdots & u_{7}\\ r_{1}\times u_{1} & r_{2}\times u_{2} & \cdots & r_{7}\times u_{7} \end{array}\right].$$

As shown in Figure 4, take a cable for analysis. $b_{i}$ represents the vector from point O to point $B_{i}$, and p represents the vector from point O to point P. According to the vector relationship of the triangle, the following equation is obtained:(7)$$L_{i}=b_{i}-p-r_{i}.$$

Deriving ***R*** with respect to time, the following is obtained:(8)$$\overset{\cdot }{R}=w\times R.$$

### 3.2. Static Stiffness Analysis

The ability of the moving platform to resist the pose change under external force is one indicator reflecting the stiffness of the mechanism [27]. Therefore, the stiffness of the mechanism can be calculated. The relationship between the external microforce d***F*** of the moving platform and the micropose d***S*** generated by the microforce is established as follows:(9)dF=K⋅dS,
where ***K*** is the static stiffness matrix of the system and $K\in R^{6\times 6}$.

From Equations (5) and (9), the following equation can be obtained:(10)$$K=\frac{dF}{dS}=-(\frac{dJ^{T}}{dS}T+J\frac{dT}{dS})=K_{T}+K_{S},$$
where ***K***_T_ is called the platform pose stiffness, and ***K***_S_ is called the cable pose stiffness. The following derivations for platform positional stiffness and cable pose stiffness were made.

The expression for the platform pose stiffness ***K***_T_ is written as:(11)$$K_{T}=-\frac{dJ^{T}}{dS}T=-WT,$$
where $W=[W^{1},W^{2},\cdots ,W^{6}]\in R^{6\times 6\times 7}$ represents the three-dimensional matrix, which can be expressed as:(12)$$W^{\left(j\right)}=\frac{dJ^{T}}{dS_{j}}=\left[\begin{array}{cccc} \frac{du_{1}}{dS_{j}} & \frac{du_{2}}{dS_{j}} & \cdots & \frac{du_{7}}{dS_{j}}\\ \frac{d(r_{1}\times u_{1})}{dS_{j}} & \frac{d(r_{2}\times u_{2})}{dS_{j}} & \cdots & \frac{d(r_{7}\times u_{7})}{dS_{j}} \end{array}\right],$$
where ***u****_i_* is the unit vector in the direction of the cable *i* vector, defined as:(13)$$u_{i}=[\begin{array}{ccc} u_{xi} & u_{yi} & u_{zi} \end{array}{]}^{T}.$$

The expression for the cable pose stiffness ***K***_S_ is expressed as:(14)$$K_{S}=J^{T}\cdot \mathrm{diag}(\frac{E_{1}\cdot A_{1}}{l_{1}},\frac{E_{2}\cdot A_{2}}{l_{2}},\cdots ,\frac{E_{7}\cdot A_{7}}{l_{7}})\cdot J,$$
where $k_{i}=E_{i}\cdot A_{i}$ is the unit stiffness of the cable, ***E**_i_* denotes the modulus of elasticity of the cable, ***A**_i_* denotes the cross-sectional area of the cable, ***l***_i_ is the length of the cable and the unit stiffness of the cable is taken as *k* = 125,000 N/m.

Derived from Equation (5), the following can be obtained:(15)$$T=-J^{+}F+\lambda (I-J^{+}J),$$
where $J^{+}$ is the M-P generalized inverse matrix of the structural matrix ***J***, and *λ* is an arbitrary constant, representing the constant value of the solution space satisfying the cable tension condition.

From Equation (15), the following can be obtained:(16)$$T_{\min}+(J^{+}F)\leq \lambda (I-J^{+}J)\leq T_{\max}+(J^{+}F),$$
where ***T***_min_ and ***T***_max_ represent the minimum tension and the maximum tension of a single cable, respectively. ***T***_min_ is 5 N, and ***T***_max_ is 200 N. The range of *λ* can be obtained from Equation (16).

## 4. Stiffness Simulation Analysis

### 4.1. Simulation Model Construction

As shown in Figure 5, the simulation model of the PCUR is built in MATLAB/Simscape Multibody. Seven groups of cable-driven units CDU1–CDU7 are connected with three traction points of the moving platform. CDU1–CDU7 are inputs with the determined torque values to make the moving platform in a static state.

The internal modeling and simulation of CDU1–CDU7 are shown in Figure 6. Starting from the World Frame, each cable-driven unit model is built to determine the motor’s position and the driving point. The end of the cable is connected with the corresponding traction point of the moving platform.

The result of the Multibody simulation model is shown in Figure 7. Seven cable-driven units are connected with the moving platform through the cable. The corresponding motion results of the moving platform are generated under the cable tension to study the stiffness characteristics of the PCUR and verify the correctness of the stiffness theory derivation.

### 4.2. λ Value Range Solution and Stiffness Variation Characteristics

The initial state of the moving platform was ***S*** = [0, 0, 0, 0, 0, 0]^T^. The generalized external force of the moving platform ***F*** was [0, 0, −10, 0, 0, 0]^T^. The center point of the moving platform coincided with the origin of the *O-xyz* spatial coordinate system. The range of *λ* was solved according to Equation (16). As shown in Figure 8, the green line represents the minimum tension of the seven cables, and the red line represents the maximum tension of the seven cables. The part sandwiched with the imaginary black line is the value range of *λ*, which was [52.57, 291.48].

According to the value range of *λ*, the relationship between the cable tension and *λ* is shown in Figure 9. As can be seen, the tensions of the 1st and 4th cables were equal, the tensions of the 5th and 7th cables were similar, and the tensions of the 2nd and 6th cables were much larger than that of the other five cables, which was determined by the configuration of the PCUR. With the increase of *λ*, the tensions of the seven cables increased accordingly, and the system stiffness also increased. Hence, *λ* also reflected the size of the system stiffness to a certain extent. In the value range of *λ*, the cables tensions were in the range of 5–200 N. That is, the cable tension can be obtained according to Figure 9.

The cable tensions under different *λ* were taken to obtain the eigenvalues of the global stiffness matrix ***K***, of which the eigenvalues represent the stiffness values in the six-DOF direction of the space, which are represented by ***K_x_***, ***K_y_***, ***K_z_***, ***K_α_***, ***K_β_***, and ***K_γ_***, respectively. Then, the relationship between the stiffness values in the six-DOF direction of the space and the *λ* can be obtained. As shown in Figure 10, it can be seen that with the increase of *λ* value, the stiffness of the system increased. The stiffness values of the moving platform ***K_x_***, ***K_y_***, ***K_z_***, and ***K_α_*** gradually increased, while ***K_β_*** and ***K_γ_*** decreased progressively. The change of the stiffness value in each DOF direction was related to the PCUR mechanism’s configuration.

## 5. Results of the Stiffness Simulation

The relationship between the system stiffness and the pose was further studied through the PCUR stiffness derivation, simulation model construction, and *λ* value range solution. Under the *λ* value range of 52.57–291.48, the vertical downward external force was applied to the moving platform in the simulation model, so that the effects of different stiffnesses on the PCUR were known.

For a given stiffness, the whole system corresponded to a spring system which maintained a stable oscillation in the elastic range. A heavy ball with a mass of 0.05 kg, suspended 200 mm from a moving platform, was in a free-fall motion. When a moving platform was shocked, it was first in an impact shock period, with an increase in the amplitude of the oscillation. Then, it was in a period of stable shocks, with relatively stable changes, as shown in Figure 11.

### 5.1. Motion of the Moving Platform with a Constant Stiffness and Different External Forces

For *λ* = 80, the tension value of each cable made the moving platform in the static state, that is, ***T*** = [8.58, 53.84, 10.31, 8.58, 9.79, 55.60, 9.79]^T^. The overall stiffness of the mechanism was calculated by Equation (17), and the overall stiffness eigenvalues were ***e*** = [2.7690, 2.0962, 0.3639, 0.0385, 0.0249, 0.0131]^T^ × 10^5^.
(17)$$K=\left[\begin{array}{cccccc} 0.3634 & 0.0356 & 0.0001 & 0.0000 & 0.0174 & 0.0046\\ 0.0356 & 2.5511 & 0.3135 & -0.0129 & -0.0407 & 0.0706\\ 0.0001 & 0.3135 & 2.3091 & 0.0000 & 0.0452 & 0.0407\\ 0.0000 & -0.0129 & 0.0000 & 0.0133 & 0.0000 & 0.0010\\ 0.0174 & -0.0407 & 0.0452 & -0.0000 & 0.0391 & -0.0053\\ 0.0046 & 0.0707 & 0.0407 & 0.0010 & -0.0053 & 0.0295 \end{array}\right]\times 10^{5}.$$

External force of 0 N (without an external force), 5 N, and 10 N were applied to the moving platform. The motion data were extracted from the mobile platform, and the motion trajectories were plotted. Figure 12 shows the position change of the center point of the moving platform without an external force. The coordinate change of the center point of the moving platform in the spatial coordinate system *O-xyz* and the linear distance curve between the center point and the initial pose were drawn (the following analysis was the same). It can be seen that the position variation of the moving platform was minimal without an external force, with a weak vibration in the 0.2 mm range. This phenomenon is that there was a slight deviation in the tension of each cable input and also by the magnitude of the gravity of the moving platform itself. In practical research, the influence is small and can be ignored.

The position change of the center point of the moving platform when the external force was 5 N is shown in Figure 13. It can be seen that the moving platform had a position change of about 200 mm under the external force. The moving platform moved due to the coupling of external forces and cables. Without considering the system damping, although the motion trajectory of the moving platform in space did not show the characteristics of the periodic motion, it only moved randomly in a particular range. Still, its linear distance showed the regular change features, and the moving platform repeatedly vibrated in a particular range.

Figure 14 shows the position change of the center point of the moving platform when the external force was 10 N. It can be seen that when an external force of 10 N was applied, the moving range of the moving platform was further increased, from 200 mm at 5 N to 370 mm. The moving platform could not withstand the external force of 10 N under lower stiffness and the occurrence of turbulence.

In summary, from Figure 12, Figure 13 and Figure 14, it can be seen that under a specific stiffness condition, the greater the external force on the moving platform, the greater the position change of the moving platform, and the instability of the moving platform gradually increased.

### 5.2. Motion of the Moving Platform with Different Stiffnesses and a Constant External Force

The following shows the position change of the moving platform under different stiffness conditions. For *λ* = 170, the tension value of each cable that kept the moving platform in a stationary state, and ***T*** was written as [20.35, 116.04, 25.86, 20.35, 18.75, 115.19, 18.75]^T^. The overall stiffness of the mechanism was obtained with Equation (18), and the eigenvalue of the overall stiffness ***e*** was expressed as [2.7692, 2.0970, 0.3649, 0.0386, 0.0249, 0.0130]^T^ × 10^5^.
(18)$$K=\left[\begin{array}{cccccc} 0.3645 & 0.0356 & 0.0002 & 0.0000 & 0.0174 & 0.0046\\ 0.0356 & 2.5517 & 0.3132 & -0.0129 & -0.0407 & 0.0705\\ 0.0002 & 0.3132 & 2.3097 & 0.0000 & 0.0453 & 0.0407\\ 0.0000 & -0.0129 & 0.0000 & 0.0132 & 0.0000 & 0.0010\\ 0.0174 & -0.0407 & 0.0452 & -0.0000 & 0.0391 & -0.0053\\ 0.0046 & 0.0708 & 0.0407 & 0.0010 & -0.0053 & 0.0294 \end{array}\right]\times 10^{5}.$$

For *λ* = 260, the tension value of each cable to keep the moving platform in a stationary state ***T*** was described as [32.11, 178.24, 41.42, 32.11, 27.72, 174.79, 27.72]^T^. The overall stiffness of the mechanism was obtained with Equation (19), and the eigenvalue of the overall stiffness ***e*** was described as [2.7695, 2.0979, 0.3660, 0.0386, 0.0248, 0.0130]^T^ × 10^5^.
(19)$$K=\left[\begin{array}{cccccc} 0.3655 & 0.0356 & 0.0002 & 0.0000 & 0.0174 & 0.0046\\ 0.0356 & 2.5522 & 0.3129 & -0.0129 & -0.0407 & 0.0703\\ 0.0002 & 0.3129 & 2.3103 & 0.0000 & 0.0454 & 0.0407\\ 0.0000 & -0.0129 & 0.0000 & 0.0132 & 0.0000 & 0.0010\\ 0.0174 & -0.0407 & 0.0452 & -0.0000 & 0.0391 & -0.0053\\ 0.0046 & 0.0708 & 0.0407 & 0.0010 & -0.0053 & 0.0293 \end{array}\right]\times 10^{5}.$$

Compared with the eigenvalues of the overall stiffness under the values of *λ* = 80, 170, and 260, the change of the overall stiffness eigenvalue conformed to the change of the stiffness shown in Figure 10.

Without an external force, the moving platform remained stationary under the action of seven cables. As shown in Figure 15, the moving platform had a weak vibration of about 0.2 mm for *λ* = 80, 170, and 260. The reason is due to the given force and the simulation deviation. It had little effect on the stiffness evaluation, which was ignored.

Figure 16 shows the position change of the moving platform under the external force of 5 N with different stiffnesses. It can be found that the distance of the moving platform from the initial pose was about 200 mm for *λ* = 80. For *λ* = 170, the distance from the initial pose of the moving platform was about 90 mm. For *λ* = 260, the distance from the initial pose of the moving platform was about 60 mm. The greater the value of *λ*, the greater the system’s stiffness, and the smaller the change in the position of the moving platform.

Figure 17 shows the position change of the moving platform under the external force of 10 N with different stiffnesses. It can be found that the position change regularity of the moving platform was the same as that under the external force of 5 N. The larger the *λ* value was, the smaller the pose variation of the moving platform was. For *λ* = 80, the distance from the initial pose of the moving platform was about 330 mm. For *λ* = 170, the distance from the initial pose of the moving platform was about 180 mm. For *λ* = 260, the distance from the initial pose of the moving platform was about 120 mm.

In summary, from Figure 15, Figure 16 and Figure 17, it can be seen that when the moving platform remained stationary under the action of the cable tension. The greater the external force applied, the greater the amount of the position change of the moving platform. The smaller the system stiffness, the more likely the moving platform will generate the pose changes. When the system’s stiffness increased, the pose change of the moving platform under the same external force conditions decreased.

Different stiffness performances were adapted to different rehabilitation training stages and training modes in the actual rehabilitation training. The training modes with different stiffnesses were selected according to the specific situation to ensure that the upper limb rehabilitation robot can achieve safety and comfort rehabilitation training.

## 6. Conclusions

In this paper, by deriving the stiffness of PCUR, the theoretical expressions of the stiffness and the calculation equations of the cable tension values were obtained. The relationships between the cable tension, the stiffness value, and the value of λ taken in the direction of each degree of freedom were analyzed. Different tension values were applied to each cable, and different external forces were applied to the moving platform under the current tension value. Then, the movement of the moving platform was observed. The following two points were analyzed:

When the stiffness of the robot system remained constant, the range of motion of the moving platform became larger as the applied external force increased. When the external force was too great for the system to withstand, the system would lose stability, and increasing the external force could increase the range of motion of the moving platform.

When the external force applied to the moving platform remained constant and the stiffness of the robot system changed, the range of motion of the moving platform decreased as the stiffness of the system increased.

Further research work will focus on two aspects:

The impedance controller is being designed to incorporate the force and the position into the same control system. The closed loop control of cable tension and the cable length variation will be achieved, indirectly controlling the system stiffness by controlling the platform pose stiffness ***K***_T_.

*λ* will be linked to the actual rehabilitation training. *λ* will be taken automatically, depending on the training patterns and indicators expected by the rehabilitator, so that rehabilitation training has different patterns of the stiffness.

## Figures and Tables

**Figure 1 micromachines-13-00253-f001:**
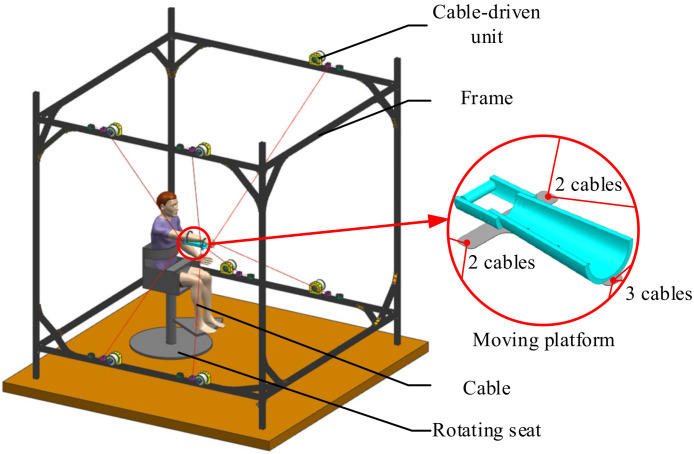
Parallel cable-driven upper limb rehabilitation robot.

**Figure 2 micromachines-13-00253-f002:**
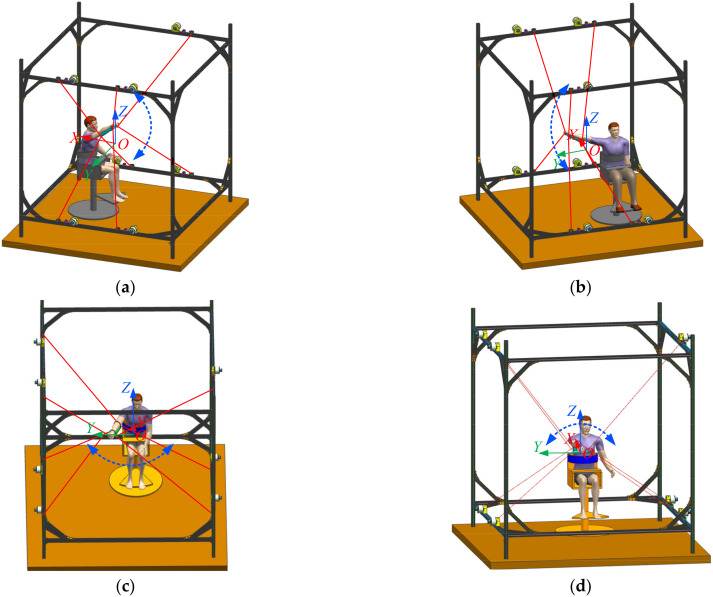
Schematic diagrams of the upper limb rehabilitation training modes: (**a**) upper limb–shoulder joint extension training; (**b**) upper limb–shoulder joint lateral extension training; (**c**) upper limb–shoulder–elbow joint extension training; (**d**) upper limb–shoulder–elbow joint internal/external rotation training.

**Figure 3 micromachines-13-00253-f003:**
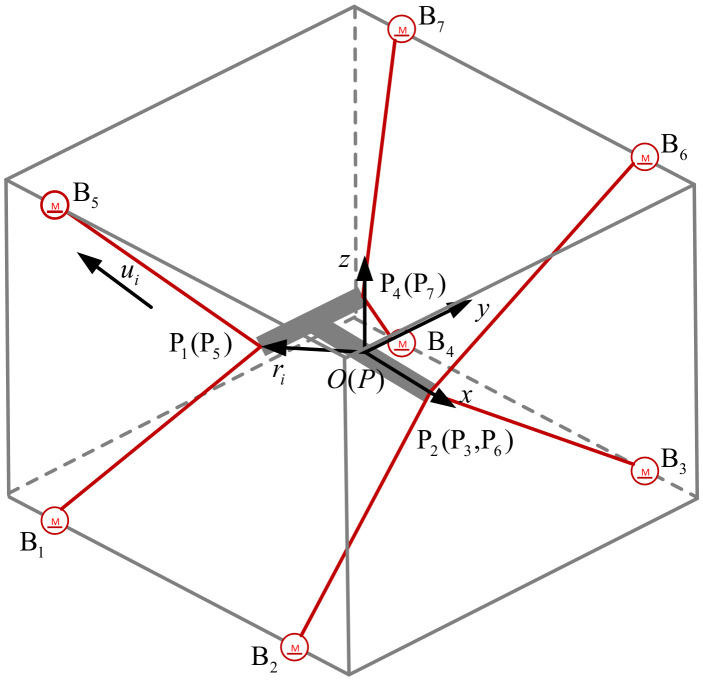
Parallel cable-driven upper limb rehabilitation robot (PCUR) mathematical modeling.

**Figure 4 micromachines-13-00253-f004:**
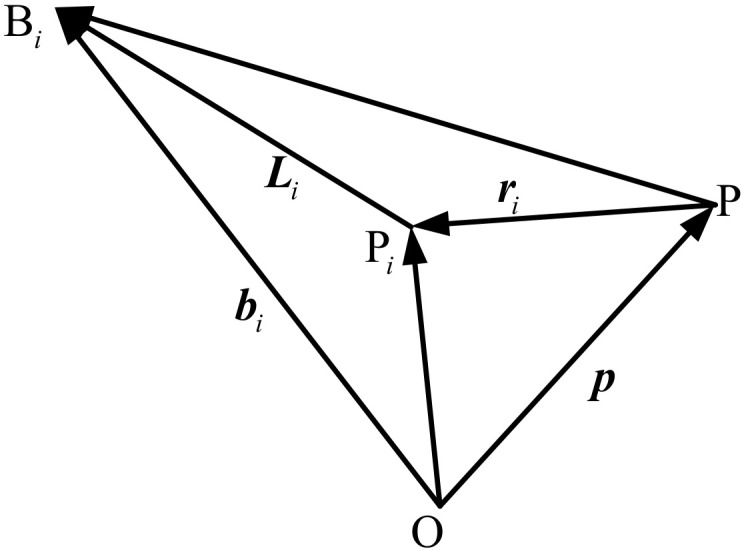
Single-cable vector schematic.

**Figure 5 micromachines-13-00253-f005:**
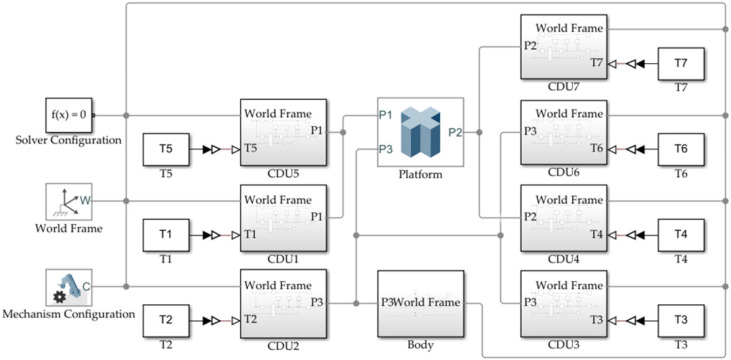
The PCUR’s stiffness analysis simulation model.

**Figure 6 micromachines-13-00253-f006:**
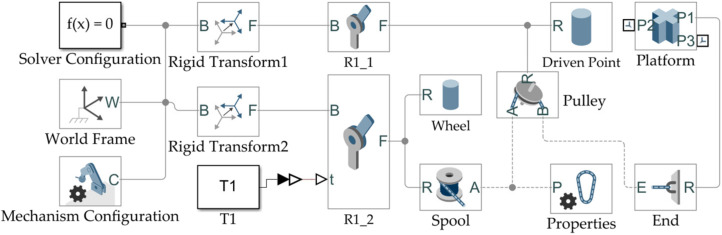
The construction of the cable-driven unit model.

**Figure 7 micromachines-13-00253-f007:**
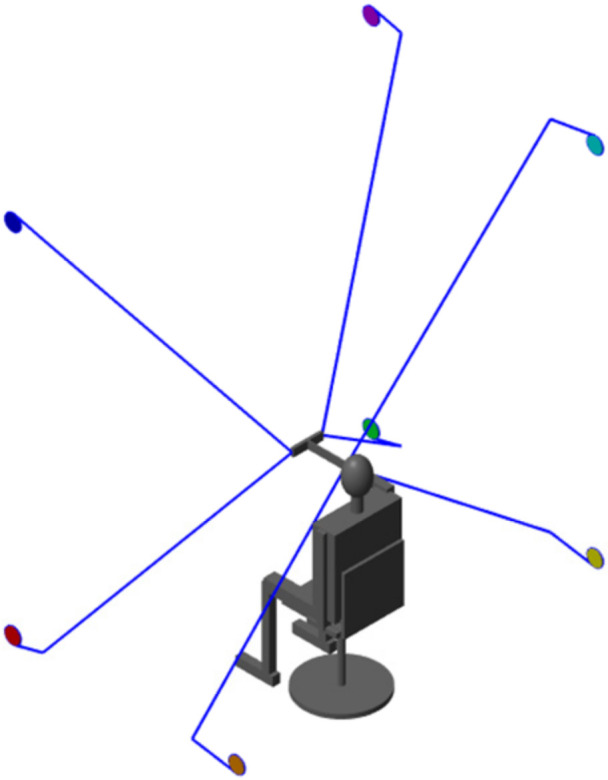
The PCUR simulation model display.

**Figure 8 micromachines-13-00253-f008:**
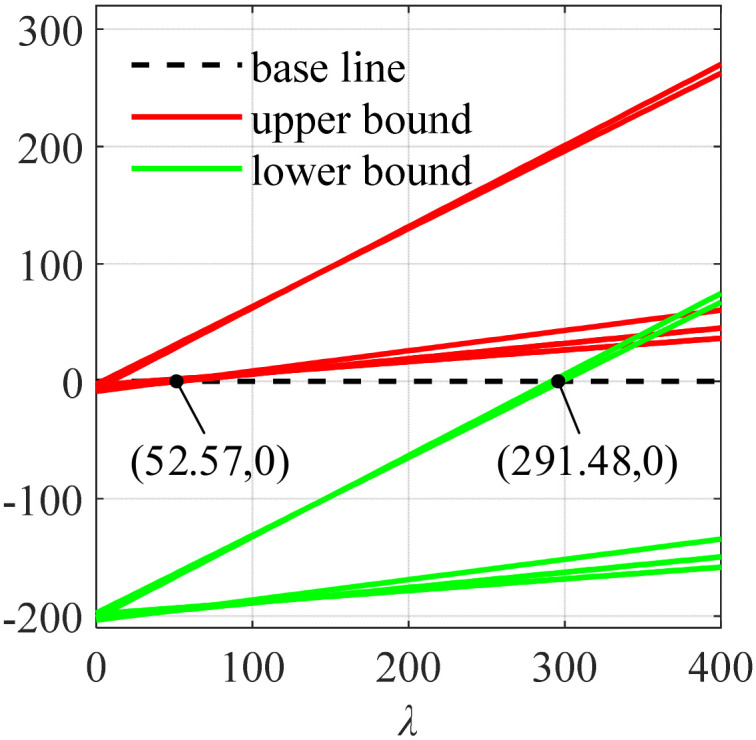
The solution of the value range of *λ*.

**Figure 9 micromachines-13-00253-f009:**
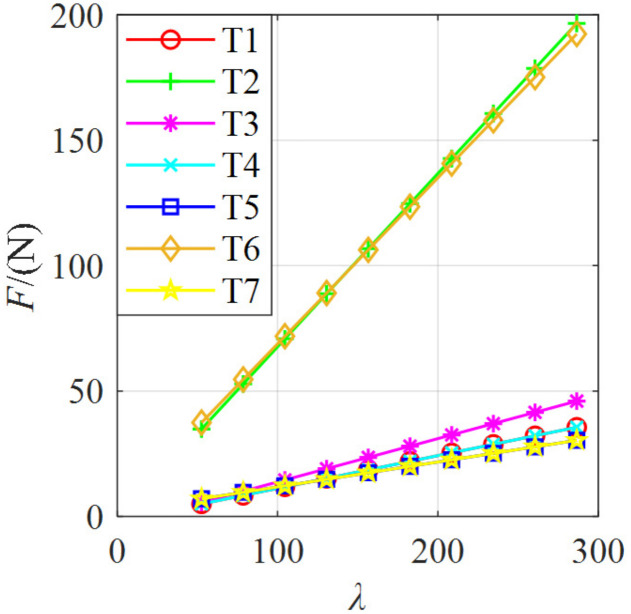
Tension changes of the seven cables within the value range of *λ*.

**Figure 10 micromachines-13-00253-f010:**
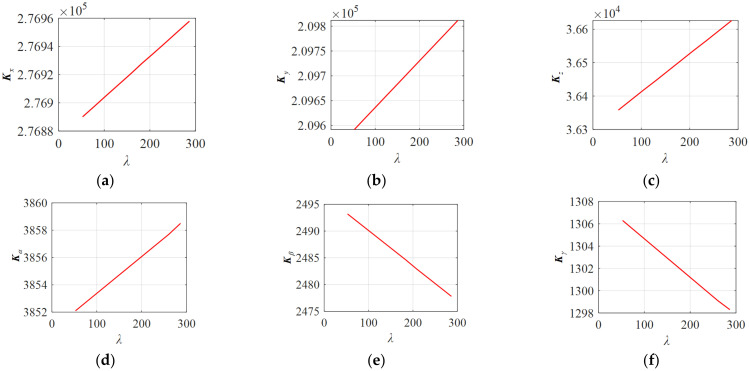
The relationship between the stiffness and *λ* in the direction of six degrees of freedom in space: (**a**) the relationship between the stiffness along the *x*-axis and *λ*, with a slope of 0.2838; (**b**) the relationship between the stiffness along the *y*-axis and *λ*, with a slope of 0.9204; (**c**) the relationship between the stiffness along the *z*-axis and *λ*, with a slope of 1.1218; (**d**) the relationship between the torsional stiffness around the *x*-axis and *λ*, with a slope of 0.0268; (**e**) the relationship between the torsional stiffness around the *y*-axis and *λ*, with a slope of −0.0645; (**f**) the relationship between the torsional stiffness around the *z*-axis and *λ*, with a slope of −0.0335.

**Figure 11 micromachines-13-00253-f011:**
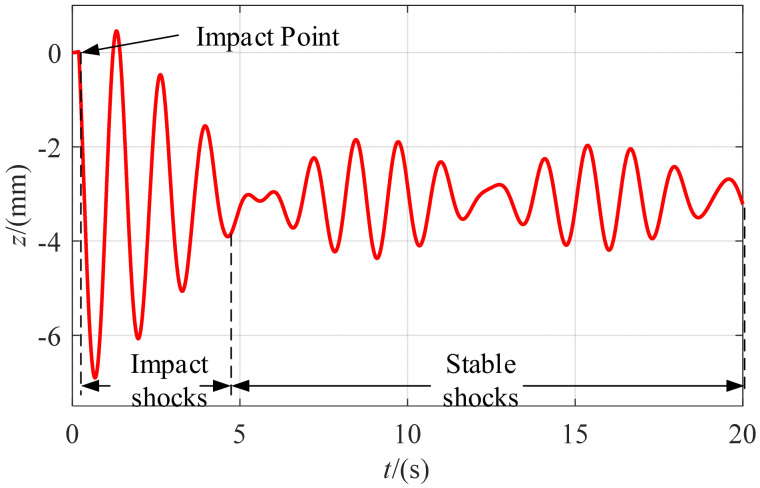
*z*-axis displacement of the moving platform by impact forces.

**Figure 12 micromachines-13-00253-f012:**
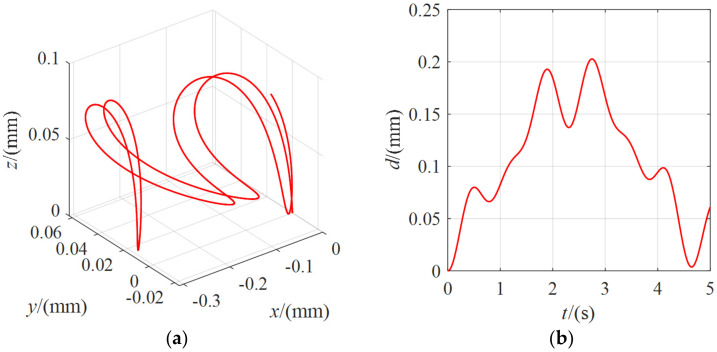
The position change of the moving platform under the external force of 0 N for *λ* = 80: (**a**) The coordinate curve of the moving platform center point in the spatial coordinate system *O-xyz*; (**b**) the variation curve of the linear distance between the center point of the moving platform and the initial pose.

**Figure 13 micromachines-13-00253-f013:**
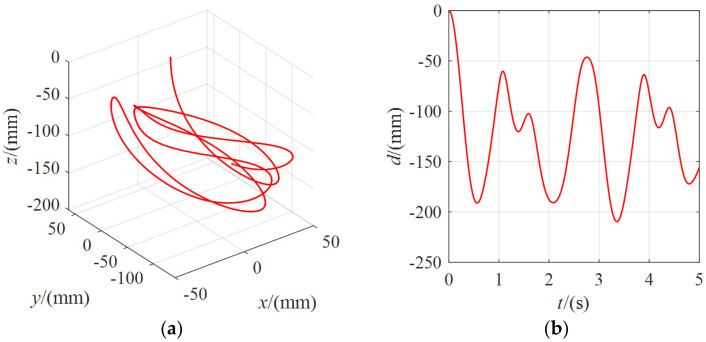
The position change of the moving platform under the external force of 5 N for *λ* = 80: (**a**) the coordinate curve of the moving platform center point in the spatial coordinate system *O-xyz*; (**b**) the variation curve of the linear distance between the center point of the moving platform and the initial pose.

**Figure 14 micromachines-13-00253-f014:**
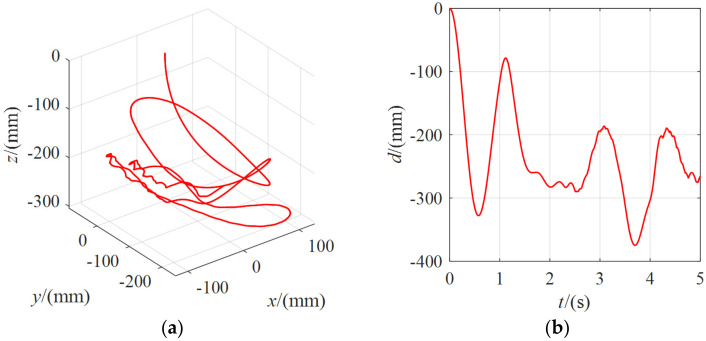
The position change of the moving platform under the external force of 10 N for *λ* = 80: (**a**) the coordinate curve of the moving platform center point in the spatial coordinate system *O-xyz*; (**b**) the variation curve of the linear distance between the center point of the moving platform and the initial pose.

**Figure 15 micromachines-13-00253-f015:**
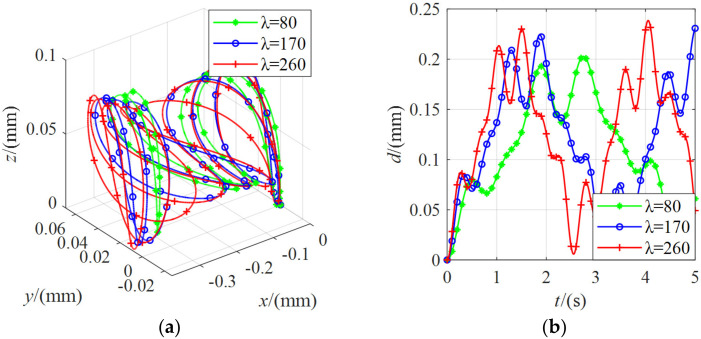
The position change of the moving platform under the external force of 0 N with different stiffnesses: (**a**) the coordinate curve of the moving platform center point in the spatial coordinate system *O-xyz*; (**b**) the variation curve of the linear distance between the center point of the moving platform and the initial pose.

**Figure 16 micromachines-13-00253-f016:**
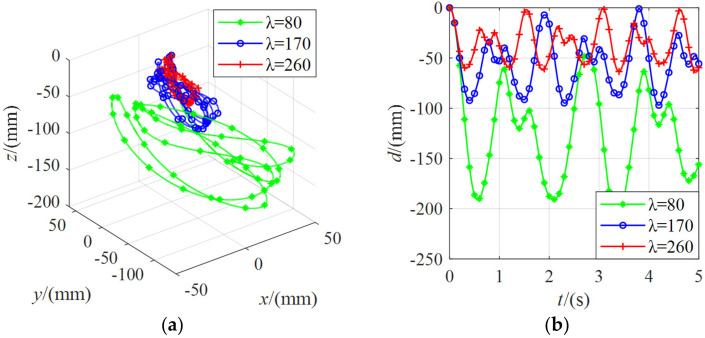
The position change of the moving platform under the external force of 5 N with different stiffnesses: (**a**) the coordinate curve of the moving platform center point in spatial coordinate system *O-xyz*; (**b**) the variation curve of the linear distance between the center point of the moving platform and the initial pose.

**Figure 17 micromachines-13-00253-f017:**
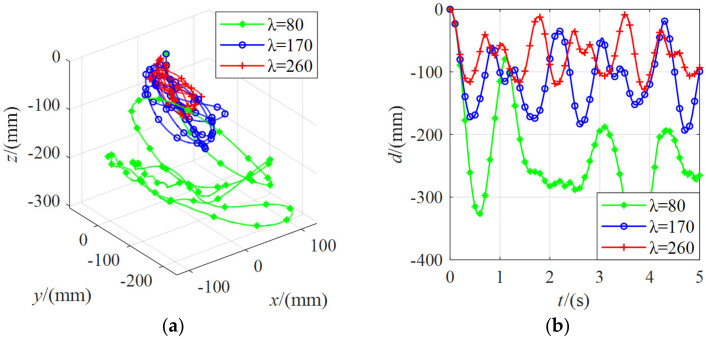
The position change of the moving platform under the external force of 10 N with different stiffnesses: (**a**) the coordinate curve of the moving platform center point in spatial coordinate system *O-xyz*; (**b**) the variation curve of the linear distance between the center point of the moving platform and the initial pose.

## Data Availability

The original data contributions presented in the study are included in the article; further inquiries can be directed to the corresponding authors.

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
