# Peer review of "Stiffness Analysis of Parallel Cable-Driven Upper Limb Rehabilitation Robot"

_micromachines, 2022, doi:10.3390/mi13020253_

Round 1

Reviewer 1 Report

Please see attached PDF

Reviewer 2 Report

This paper shows the stiffness analysis of CDRP in upper limb rehabilitation applications. The main issues are as follows:

1. Lack of innovation. Most equations ((1)-(25)) have been presented by other publications. The authors should carefully cite their work.
2. The authors are suggested to highlight their contributions in the introduction.
3. Line 203: where is "G" in the equation(s)?
4. In Fig. 8, please change the position of the legend.
5. Other suggestions: the authors are suggested introducing disturbances, etc. to improve their theoretical work and conduct experimental validations.

Reviewer 3 Report

The subject of the paper is of great interest in developing original cable-driven upper limb rehabilitation robots. The paper represents a good work and yours experience and results might stimulate discussion regarding this topic. Please Consider following suggestions:

- please emphasize elements of novelty and originality of this paper, specify the original contributions of this paper, in relation to similar (yours) papers in the same field;

- please extend some of yours conclusions;

- although you mentioned one sentence about your  future work in this area, please add comments and give more details.

Round 2

Reviewer 1 Report

see attached PDF
